# In the Search for Novel, Isoflavone-Rich Functional Foods—Comparative Studies of Four Clover Species Sprouts and Their Chemopreventive Potential for Breast and Prostate Cancer

**DOI:** 10.3390/ph15070806

**Published:** 2022-06-28

**Authors:** Agnieszka Galanty, Monika Niepsuj, Marta Grudzińska, Paweł Zagrodzki, Irma Podolak, Paweł Paśko

**Affiliations:** 1Department of Pharmacognosy, Faculty of Pharmacy, Jagiellonian University Medical College, Medyczna 9, 30-688 Cracow, Poland; monika.niepsuj13@gmail.com (M.N.); grrudzinska@gmail.com (M.G.); irma.podolak@uj.edu.pl (I.P.); 2Department of Food Chemistry and Nutrition, Faculty of Pharmacy, Jagiellonian University Medical College, Medyczna 9, 30-688 Cracow, Poland; pawel.zagrodzki@uj.edu.pl (P.Z.); p.pasko@uj.edu.pl (P.P.)

**Keywords:** clover sprouts, *Trifolium*, isoflavones, prostate cancer, breast cancer, cytotoxic

## Abstract

Despite a significant amount of research, the relationship between a diet rich in isoflavones and breast and prostate cancer risk is still ambiguous. The purpose of the current study was to pre-select the potential candidate for functional foods among red, white, crimson, and Persian clover sprouts, cultured for different periods of time (up to 10 days), with respect to the isoflavone content (determined by HPLC-UV-VIS), and to verify their impact on hormone-dependent cancers in vitro. The red clover sprouts were the richest in isoflavones (up to 426.2 mg/100 g dw), whereas the lowest content was observed for the crimson clover. Each species produced isoflavones in different patterns, which refer to the germination time. Hormone-insensitive MDA-MB-231 breast cancer cells were more resistant to the tested extracts than estrogen-dependent MCF7 breast cancer cells. Regarding prostate cancer, androgen-dependent LNCap cells were most susceptible to the tested sprouts, followed by androgen-insensitive, high metastatic PC3, and low metastatic DU145 cells. The observed cytotoxic impact of the tested sprouts is not associated with isoflavone content, as confirmed by chemometric analysis. Furthermore, the sprouts tested revealed a high antioxidant potential, and were characterized by high safety for normal breast and prostate cells.

## 1. Introduction

Plants from the *Fabaceae* family, as well as their sprouts and by-products, are an attractive object of research which are currently gaining more and more scientific interest. This is mainly due to their nutritional value, but also their health-promoting potential, associated with the presence of polyphenolic compounds [1], among which, isoflavones, a class of so-called phytoestrogens, are the most intriguing. As they are structurally similar to 17-β-estradiol, isoflavones reveal estrogenic activity, both by affecting estrogen receptors and aromatase activity (which regulates transformation of androstenedione into estrone). Furthermore, isoflavones can also affect 5-α-reductase activity, which is responsible for the transformation of testosterone to dihydrotestosterone. This may be of great importance in terms of the development of hormone-sensitive cancers, including breast and prostate cancers, as these compounds can compete with 17-β-estradiol for the binding site of estrogen receptors. Literature data indicate that isoflavones trigger epigenomic effects that could be beneficial in breast cancer prevention and/or treatment [2], and they can reduce the risk of inducing estrogen-dependent tumorigenesis [3]. However, some studies point out that at low concentrations, isoflavones may stimulate the growth of breast cancer cells [4]. Regarding prostate cancer, isoflavones, in addition to affecting the estrogen receptor, also modulate the androgen receptor, resulting in a decreased risk of the carcinogenesis process, but the detailed signalling pathways are still unknown [5]. Despite a significant amount of research, the relationship between a diet rich in isoflavones and breast and prostate cancers is still ambiguous and controversial [6,7,8].

Functional food, in addition to its nutritional effect, reduces the risk of various diseases, including cancer. Sprouts are an attractive and dynamically developing example of functional food, with benefits such as high nutritional and low caloric values. Additionally, during the germination process, the level of active compounds can increase significantly, as was demonstrated especially for phenolic acids and flavonoids [9], but also for isoflavones [10]. Sprouts rich in phytoestrogens (including isoflavones) are an example of functional food that may reveal chemopreventive potential against the development of hormone-dependent cancers. However, as the isoflavones provided with the diet can influence the hormonal balance in the human body, it is crucial to know what their content in a given product is. This is especially important for people with the risk of the development or already-existing hormone-dependent cancers. During the search for novel dietary sources of isoflavones, we focused on clover (*Trifolium*) species, belonging to the *Fabaceae* family, of which, red clover (*T. pratense* L.) is the most well-known. In addition to soy, red clover is one of the richest sources of isoflavones, and its aerial parts are used as dietary supplements to reduce menopausal symptoms. Only a few studies described the level of these compounds in red clover sprouts, and the results are very promising [10,11,12], although significant differences were also observed between particular studies. The results also indicate the impact of red clover and its sprouts on breast cancer cells and their estrogenic activity in vitro [13,14,15]. Therefore, we decided to include red clover sprouts in our study as an example of the most-known clover species, and to compare them with the sprouts of three other clovers, namely, white (*T. repens* L.), crimson (*T. incarnatum* L.), and Persian (*T. resupinatum* L.), in terms of the isoflavone profile and the impact on breast and prostate cells. Interestingly, except for the latter, these clover species have never been used to grow sprouts. As far as Persian clover sprouts are concerned, the results on their antioxidant properties have been published [16,17], but no information on the content of isoflavones can be found.

The purpose of this study was to preselect the potential candidate for functional foods from among the four clover sprouts mentioned above, cultured for different periods of time, with respect to the isoflavone content and the importance in the chemoprevention of hormone-dependent cancers. To achieve the assumed goal, except from HPLC analysis, we used a specially designed in vitro cellular model, comprising breast and prostate cancer cells, differing in their hormone sensitivity. Normal breast and prostate cells were also included in the study to determine the safety of the examined clover sprouts. To better justify our results and reveal potential relationships between the studied factors, chemometric analysis was applied.

## 2. Results and Discussion

This study aimed at the comparative analysis of four clover species sprouts, in the context of the search for new candidates for functional food with chemopreventive potential in hormone-dependent breast and prostate cancers. We used three clover species and compared them to red clover sprouts, which are a popular food product. It is also worth noting that this is the first study to describe the sprouts of white and crimson clover, and also to define the isoflavone content of Persian clover.

### 2.1. Four Clover Species Sprouts Differ in Their Isoflavones Profile

In the first step of the experiment, the qualitative profile of isoflavones and the quantitative accumulation dynamics in the sprouts grown for 3, 5, 7, and 10 days were determined by HPLC, and the results are presented in Table 1. Two isoflavone glycosides, namely, ononin and daidzin, together with three free isoflavones: genistein, daidzein, and formononetin, were identified in the examined sprout extracts. Most of the seed extracts contained only trace amounts of isoflavones, apart from PCS and CCS, where genistein, daidzein, and formononetin were detected in small amounts. None of the tested species revealed the presence of biochanin A or genistin. Notable qualitative differences were observed between the four clover species. The PC and RC sprouts contained five (genistein, daidzein, ononin, formononetin, daidzin) and four (genistein, ononin, formononetin, daidzin) of the examined isoflavones, respectively, whereas only two isoflavones were observed in the WC (ononin, formononetin) and CC (genistein, formononetin) sprouts. The isoflavone profile for RC sprouts is consistent with the observations of other authors [11]; however, some of them also indicated the presence of biochanin A, genistin, sissotrin, or glycytein [10,12]. No information has been published so far on the isoflavone profile in WC, PC, and CC sprouts; thus, our analysis is the first one concerning their presence.

In terms of quantitative analysis, the absolute winner in isoflavone content among the tested species is RC, with the sprouts containing between 204.1 and 426.2 mg/100 g dw of isoflavones sum (Figure 1). The WC and PC sprouts contained 12.92–48.12 and 0.26–47.06 mg/100 g dw of isoflavone sum, respectively, whereas their sum in the CC sprouts was the lowest (0.07–23.31 mg/100 g dw). Ononin and formononetin were present in the highest amount in the examined sprouts. Interestingly, ononin was the predominant isoflavone in RC and PC sprouts, whereas formononetin was predominant in WC and CC sprouts (Table 1). Similar observations were made by [10] for red clover sprouts; however, the total isoflavone content was twice as much as our results, due to the higher amount of formononetin (up to 215.55 mg/100 g dw for 10-day sprouts). The contents of ononin, daidzin, and genistein were comparable to those described by [10] for ten-day sprouts, whereas [12] noted a significantly lower amount of formononetin in 5-day red clover sprouts compared to our study. Such a difference may result from the details in the growth conditions, or the variety used for the experiment.

### 2.2. Different Pattern in Isoflavones Accumulation Dynamics among the Tested Sprouts

The accumulation dynamics of isoflavones revealed different patterns in their synthesis among the tested sprouts. In RC sprouts, the content of ononin and genistein was constantly increasing with the sprouting time, whereas formononetin and daidzin were produced in an increase and fall mode, but with different maximum content (in RC5 and RC10 sprouts, respectively). In WC sprouts, the content of ononin and formononetin was increasing up to the seventh day, followed by a drastic decrease on the last day of sprouting. The PC3 sprouts had the highest content of all isoflavones examined, with a further decrease, especially observed on the last day of culturing. In the case of CC sprouts, two different patterns of isoflavone accumulation were observed: the rise and fall mode for genistein, with its highest content in PC10, and the drastic increase followed by a constant decrease for formononetin, with its highest amount noted for PC3 sprouts. Our results are just opposite to the accumulation dynamics patterns in red clover sprouts described by [10], with maximum content of formononetin, daidzin, and genistein in 10-day sprouts, and ononin in 7 day-sprouts. Another study noted a constant decrease in daidzin synthesis in red clover sprouts cultured for 5 days, whereas the amount of formononetin and genistein increased with the time of culture, with the maximum in the fourth and third day, respectively, followed by a decrease [12]. The results obtained, compared to those obtained by other authors, suggest considerable fluctuations of isoflavone amounts and accumulation in clover sprouts, as a result of the details in the culturing and analytical conditions.

### 2.3. Comparison of Cytotoxic Potential of Four Clover Species Sprouts

In the next step of the experiment, we focused on the cytotoxic potential of the tested sprouts which varied in isoflavone content. Taking into account the estrogenic properties of isoflavones, as a cellular model, we have chosen hormone-dependent and independent cancer cells, namely, breast and prostate. The model was also completed with appropriate normal breast and prostate cells to verify the safety of the tested samples.

#### 2.3.1. Impact of the Tested Sprouts on Viability of Breast Cancer and Normal Cells

To evaluate the cytotoxic impact of the sprouts tested, we used two breast cancer cell lines, differing not only in metastatic potential, but also in the expression of hormone receptors: low invasive, estrogen- and progesterone-receptor positive MCF7, and high metastatic, estrogen- and progesterone-receptor negative MDA-MB-231 cells. Normal epithelial breast MCF10A cells were also included in the experimental model. The results are presented in Table 2 as IC_50_ values, and Figure 2 (for the highest concentration tested of 500 µg/mL). The highly invasive MDA-MB-231 cells were found to be less susceptible to the tested extracts compared to MCF7 cells, with the exception of the CC10 sprouts (IC_50_ 56.7 and 61.1 µg/mL, respectively). However, for most samples, the IC_50_ values exceeded the maximum concentration used in the assay. The highest cytotoxic effect against MCF7 cells was observed for CC7 and CC10, but also PC5 and PC10 sprouts, and the differences between the IC_50_ values for these samples were not statistically significant. The most important thing is that at the concentrations cytotoxic to cancer cells, the sprouts tested were characterized by high safety for normal breast cells, and the decrease in their viability was observed only at the higher concentrations tested (300–500 µg/mL). The most interesting observation from our study comes from the relationship between the isoflavone content and the cytotoxic potential of the examined sprouts. Sprouts rich in isoflavones were almost inactive in breast cancer cells, whereas CC sprouts, with the lowest amount of isoflavones, expressed a high cytotoxic impact and were safe for normal cells at the same time.

Recently, red clover sprouts were reported to influence MCF7 and MDA-MB-231 cells by [15], and the IC_50_ values were 15 mg of isoflavones in dw of sprouts/mL, which is a much weaker effect than that obtained in our study. The estrogenic potential of commercial red clover sprouts was also observed, as a stimulation of MCF7 cell proliferation [13]. Our results for WC, CC, and PC sprouts described their cytotoxic activity to breast cancer cells for the first time. The observed cytotoxic impact of the tested sprouts, not associated with isoflavones, may be due to the presence of other compounds, such as chlorogenic or gallic acid, which were also present in noticeable amounts in the extracts (data not shown). Some recent studies demonstrated the interesting cytotoxic potential of the two compounds in breast cancer cells [18,19,20,21], but further analysis is needed to prove this relationship.

#### 2.3.2. Impact of the Tested Sprouts on Viability of Prostate Cancer and Normal Cells

As information on the impact of isoflavones on prostate cancer cells is scarce, and its results are ambiguous, we decided to verify the effect of the examined clover sprouts, differing in isoflavone content, on androgen-dependent LNCap, and androgen-insensitive prostate cancer cells, DU145 and PC3, with low and high metastatic potential, respectively. The model was also completed with normal prostate epithelial cells, PNT2. The results are presented in Table 2 as IC_50_ values, and Figure 3 (for the highest concentration tested of 500 µg/mL). Interestingly, androgen-dependent LNCap cells were most susceptible to the tested sprouts, followed by high metastatic but androgen-insensitive PC3 cells, whereas low metastatic DU145 cells were most resistant. Similar to the effect obtained for breast cancer cells, CC and PC sprouts, with low amounts of isoflavones, revealed the highest cytotoxic impact to the examined cancer cells. Interestingly, the effect observed for normal prostate cells was very weak, with the lowest IC_50_ of 119.5 and 237.9 µg/mL only for CC10 and CC7 sprouts, respectively, which suggests that at the doses cytotoxic to cancer cells, the tested extracts (with the exception of CC10 and CC7 samples) were safe to normal cells. The observed cytotoxicity of four clover species sprouts in prostate cancer and normal cells is probably the first attempt to describe such an effect for any clover species. However, again, the cytotoxic impact of the extracts tested did not correlate with the isoflavone content. Based on some published studies so far [22,23,24], the observed effect may also be associated with the presence of chlorogenic and gallic acid, but this speculation needs further examination.

### 2.4. Antioxidant Potential of Clover Sprouts Not Always Corresponds with Isoflavones Amount

Isoflavones are not only known for their estrogenic activity, but because they belong to the class of polyphenolics, they also reveal antioxidant properties [25]. Oxidative damage is often related to the initiation and progression of carcinogenesis [26]. Moreover, some antioxidants can also protect normal tissues from chemo- or radiotherapy side effects [27]. Therefore, in the last step of the experiment, we decided to verify and compare the antioxidant potential of clover sprouts tested by the FRAP and DPPH assay, and the results are presented in Table 1. RC and PC sprouts revealed the strongest antioxidant potency measured by FRAP, whereas in the DPPH assay, WC sprouts achieved the highest results. Interestingly, for each species examined, the highest results were observed for both the FRAP and DPPH assays for 5- and 10-day sprouts. The activity was not dependent on the content of isoflavones in the sprouts, apart from the RC sprouts, where daidzin (CW = 0.154), ononin (CW = 0.144), and formononetin (CW = 0.132) were positively correlated with the antioxidant activity evaluated by the FRAP method. In a similar study, 2-day red clover sprouts revealed the highest antioxidant activity, measured by DPPH, followed by a constant decrease in activity up to the last, fifth day of culturing. In contrast to our results for RC sprouts, the antioxidant potential did not correspond to total isoflavones [28]. Two studies described the antioxidant potential of PC sprouts; however, the results are presented as % of activity, and thus, it is hard to compare with our observations [16,17]. In the latter study, the sprouts were grown for 4 days without light, and the antioxidant activity increased during the culturing time, which is opposite to the pattern observed in our study, probably due to the different light conditions. In the study of [17], the examination was performed on commercial sprouts of unknown age. It is worth highlighting that the antioxidant potential of WC and CC sprouts was demonstrated for the first time in our study.

### 2.5. Chemometric Analysis Reveals Some Relationships between the Studied Factors

The statistically significant PCA model of two significant components was derived, with eigenvalues of 4.05 and 3.23, respectively. The model explained 72.8% of the variance of the original parameters and the results are presented in Figure 4 and Figure 5. The first principal component in this model had predominantly negative weights for the original variables. Among them were: ononin, formononetin, daidzin, genistein, and FRAP. Therefore, all the compounds mentioned had high correlation weights with the analyzed antioxidant status index. They were divided into two clusters containing three first, and two last parameters, respectively. The second principal component was loaded, mainly positively, by MCF7 and three other mutually correlated parameters, PNT2, MCF10A, and MDA-MB-231, which were in one tight cluster (Figure 4). In particular, the highest positive correlation weights based on this component, with similar values, were disclosed between PNT2, MCF7, and MCF10A. This confirmed that both breast cancer cell lines, MDA-MB-231 and MCF7, were almost unaffected by the sprouts tested, to a similar degree as normal breast and prostate cells. There were no negative correlations between these parameters and any other parameters (Appendix A). The parameter clearly different from the others, in both principal components, was LNCaP. This can be explained due to its androgen sensitivity, which is a distinguishing feature from the other two prostate cancer cell lines used in the study.

## 3. Materials and Methods

### 3.1. Reagents

Dimethyl sulfoxide (DMSO), chloroform, HPLC grade acetonitrile, water, and formic acid were purchased from Sigma-Aldrich (Seelze, Germany). Reference standards for HPLC analysis of isoflavones: genistein, genistin, glycytein, daidzein, daidzin, ononin, biochanin A, and formononetin were purchased from Fluka Chemie. Trolox (6-hydroxy-2,5,7,8,-tetramethyl-chroman-2-carboxylic acid); and FeCl_3_·6H_2_O; 1,1-diphenyl-2-picrylhydrazyl (DPPH) were from Sigma Chemical Co. (St. Louis, MO, USA). 2,4,6-Trispyridyl-s-triazine (TPTZ) was purchased from Fluka Chemie (Buchs, Switzerland). Methanol, acetic acid, ammonium hydroxide solution, hydrochloric acid, sodium acetate, and sodium carbonate were from Avantor Performance Materials Poland S.A. (Gliwice, Poland). All reagents were of analytical grade. Distilled water was purchased from Sigma-Aldrich.

### 3.2. Plant Material

The seeds of four clover species, namely, red clover (*Trifolium pratense* L., Nike variety), white clover (*T. repens* L., Grasslands variety), crimson clover (*T. incarnatum* L., Opolska variety), and Persian clover (*T. resupinatum* L., Celtico variety), obtained from plants cultivated in Poland, were purchased in Małopolska Hodowla Roślin (Kraków, Poland). Voucher specimens (KFg/2021/T1p; KFg/2021/T2r; KFg/2021/T3i; KFg/2021/T4rs) were placed in the Department of Pharmacognosy Jagiellonian University Medical College. The seeds were transferred to EQMM Easy Green Microfarm, and grown for 3, 5, 7, and 10 days after seeding, at 25 ± 2 °C, 70% humidity, in sunlight exposure (10 h/day), being watered 3 times a day. For the purpose of this manuscript, the obtained samples are denoted RC for red clover, WC for white clover, CC for crimson clover, and PC for Persian clover, with appropriate numbers indicating the cultivation period. The samples prepared from the seeds are denoted with the letter S, added to the acronym of the clover species.

### 3.3. Preparation of Extracts and Quantitative Analysis

The plant material was Soxhlet extracted with methanol, as previously described [29]. Extracts obtained were decanted, centrifuged, and stored in darkness in a freezer at −20 °C prior to analysis for the quantity and antioxidant capacity of isoflavones. The methanol extracts were further evaporated, and the dry residues were dissolved in DMSO and used for the determination of cytotoxicity.

The quantitative analysis of isoflavones in clover sprouts was performed, as previously described [29], on the Dionex HPLC system, equipped with a PDA 100 UV-VIS detector and a Hypersil Gold (C-18) column (5 μm, 250 × 4.6 mm, Thermo EC). Analysis was carried out in gradient mode, with 1% formic acid in water (A) and acetonitrile (B), 5–60% B in 60 min, at a flow rate of 1 mL/min. The detection wavelengths used were 254 and 285 nm. The compounds mentioned above were identified by comparing their retention times with those of the reference standards. The isoflavone content was calculated by measuring the peak area with respect to the appropriate standard curve (concentration range 0.0625–1 mg/mL). All analyses were performed in three independent experiments, and the mean value was expressed in mg/100 g of dw.

### 3.4. Determination of the Antioxidant Capacities

The analysis was performed using the DPPH and FRAP methods, as previously described [29]. Briefly, DPPH methanolic solution (3.9 mL, 25 mg/L) was mixed with clover sprout extracts (0.1 mL). The reaction was monitored at 515 nm until the absorbance was constant. Each sample was measured in three replicates. The mean capacity was expressed as µM Trolox/g dw. Fresh, working FRAP solution (900 μL; 2.5 mL 10 mM ferric-tripiridyltriazine in 40 mM HCl, 2.5 mL 20 mM FeCl_3_·H_2_O and 25 mL 0.3 mol/L acetate buffer, pH 3.6) was mixed with 90 μL of distilled water and 30 μL of clover sprout extracts, and measured at 593 nm. All analyses were performed in three independent experiments. The mean capacity was expressed as µM Fe^2+^/g dw. The absorbance of both antioxidant assays was measured using a Biotek Synergy microplate reader (BioTek Instruments Inc., Winooski, VT, USA).

### 3.5. Cell Cultures and Viability Assay

Cytotoxic activity was tested on two panels of human cancer and normal cells: prostate panel (androgen-insensitive prostate carcinoma DU-145, derived from the metastatic site: brain, ATCC HTB-81; androgen-insensitive, grade IV prostate carcinoma, PC-3, derived from the metastatic site: bone, ATCC CRL-1435; androgen-sensitive prostate adenocarcinoma LNCaP, derived from the metastatic site: lymph node, ATCC CRL-1740; prostate epithelial cells, PNT2, ECACC 95012613), and breast panel (ER-positive breast adenocarcinoma MCF7, ATCC HTB-22; ER-negative breast adenocarcinoma MDA-MB-231, ATCC HTB-26; breast epithelial MCF10A, ATCC CRL-10317). Cells were grown under standard conditions (37 °C, 5% CO2, relative humidity) and culture media (DMEM/F12 for PNT2, PC3, MDA-MB-231; DMEM Low Glucose for DU145; RPMI1640 with sodium pyruvate for LNCaP; MEM with NEAA for MCF7; DMEM/F12 with 20 ng/mL epidermal growth factor (EGF), 10 µg/mL insulin, 0.5 µg/mL hydrocortisone, 100 ng/mL cholera toxin), supplemented with 10% fetal bovine serum (FBS) or 5% donor horse serum for MCF10A, and 1% antibiotics solution (10, 000 U penicillin and 10 mg streptomycin/mL). The stock solutions of the examined extracts, prepared in DMSO, were then diluted in the culture medium to the working concentrations (from 0 to 500 μg/mL). Cell viability was determined after 24 h of incubation by MTT assay, as previously described [30]. The absorbance was measured at 570 nm using a Biotek Synergy microplate reader (BioTek Instruments Inc., Winooski, VT, USA). All analyses were performed in three independent experiments, and the results are expressed as % of cell viability (mean ± SD) and IC_50_ values (concentration at which viability is inhibited by 50 percent).

### 3.6. Statistical Analysis

Statistical analysis was performed using Statistica v.13 (Statsoft, Tulsa, OK, USA). The results obtained were analyzed using one-way analysis of variance (ANOVA) followed by a post hoc Tukey’s test. All experiments were carried out in triplicate, and the data were reported as the mean ± standard deviation (SD). Furthermore, the differences between the groups were considered statistically significant when the *p*-values were 0.05 or less.

The principal component analysis (PCA) model was used to describe the correlation structure between the parameters. Parameters with large loadings on the first two principal components (>0.3) were assumed to be correlated with each other. To express the strength of bivariate associations, for pairs of correlated parameters, the algebraic products of their corresponding loadings and the cosine of the corresponding angle were calculated (these coefficients are called the correlation weights). The “corresponding angle” means the angle determined by the two lines connecting the origin with the coordinates of both parameters on the PCA loadings plot. The PCA approach was also applied to check whether the clusters of various species of clover sprouts appear in the PCA score plot. Statistical analyses were performed using STATISTICA v. 13.3. package (TIBCO Software Inc., Palo Alto, CA, USA) and SIMCA-P v.9 (Umetrics, Umeå, Sweden). The software provided by MP System Co. (Chrzanów, Poland) was used to calculate correlation weights for the pairs of parameters in the PCA model.

## 4. Conclusions

The *Trifolium* genus can be an interesting source for new candidates for functional food, as the sum of isoflavones in red clover sprouts (up to 426.2 mg/100 g dw) is comparable to what is described for soy (150–450 mg/100 g dw), and significantly higher than commercially available alfalfa sprouts (180 mg/100 g dw) [31,32]. The sum of isoflavones calculated for fresh red clover sprouts is approximately 20 mg/100 g, and this amount is comparable to some dietary supplements used for menopause. The time of sprouting had various effects on the concentration of isoflavones, depending on the species. The observed cytotoxic effect on breast and prostate hormone sensitive cancer cells was associated with a low amount of isoflavones, as it was especially observed for crimson and Persian clover sprouts. Importantly, all of the sprouts tested were safe for normal breast and prostate cells.

Our results indicate the need to implement some optimization and/or standardization procedures in the culture of clover sprouts to obtain batches with a more stable/defined level of these compounds, and a more probable prediction of their biological impact. Thus, special caution should be undertaken before including clover sprouts in the diet of consumers with the risk of hormone-dependent cancers. Moreover, women taking isoflavone-rich dietary supplements for reducing their menopausal symptoms should also be aware that some clover sprouts can deliver an additional amount of isoflavones. These observations may be also important for functional food producers. More in-depth studies are needed on the influence of isoflavone-rich sprouts on other aspects of cell functioning.

## Figures and Tables

**Figure 1 pharmaceuticals-15-00806-f001:**
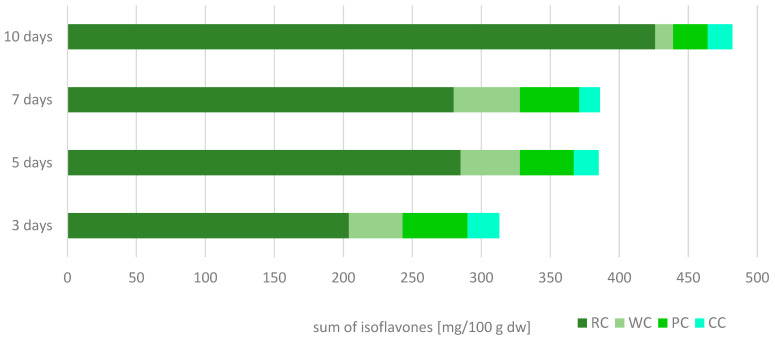
Cumulation dynamics of isoflavones sum in red (RC), white (WC), Persian (PC), and crimson (CC) clover sprouts harvested for 3, 5, 7, and 10 days.

**Figure 2 pharmaceuticals-15-00806-f002:**
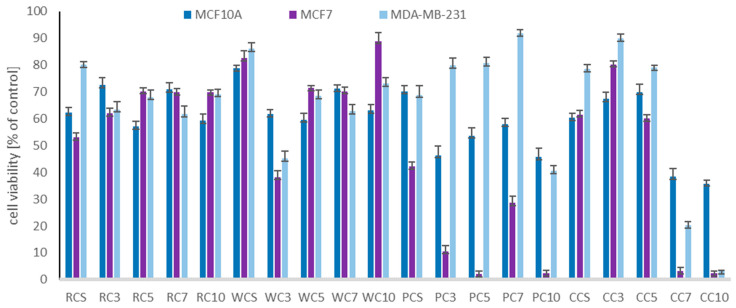
Cytotoxic effect of the extracts of red (RC), white (WC), Persian (PC), and crimson (CC) clover seeds (S) and sprouts harvested for 3, 5, 7, and 10 days to breast normal (MCF10A) and cancer (MCF7, MDA-MB-231) cells. Cells were treated with 500 µg/mL of sprout extracts (*n* = 3) for 24 h. Values are presented as the mean ± SD (standard deviation). Significant differences are shown in Appendix A.

**Figure 3 pharmaceuticals-15-00806-f003:**
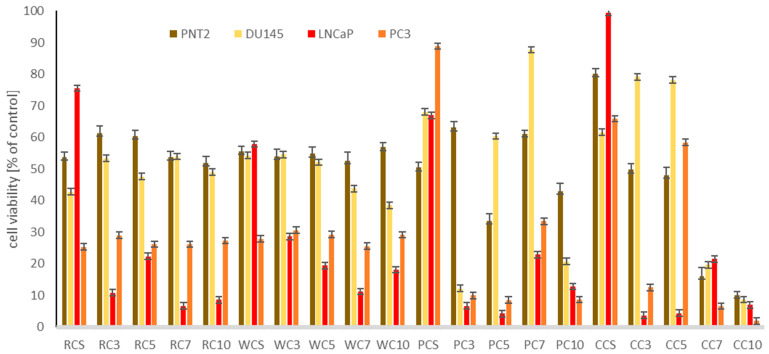
Cytotoxic effect of the extracts of red (RC), white (WC), Persian (PC), and crimson (CC) clover seeds (S) and sprouts harvested for 3, 5, 7, and 10 days to prostate normal (PNT2) and cancer (DU145, PC3, LNCaP) cells. Cells were treated with 500 µg/mL of sprout extracts (*n* = 3) for 24 h. Values are presented as the mean ± SD (standard deviation). Significant differences are shown in Appendix A.

**Figure 4 pharmaceuticals-15-00806-f004:**
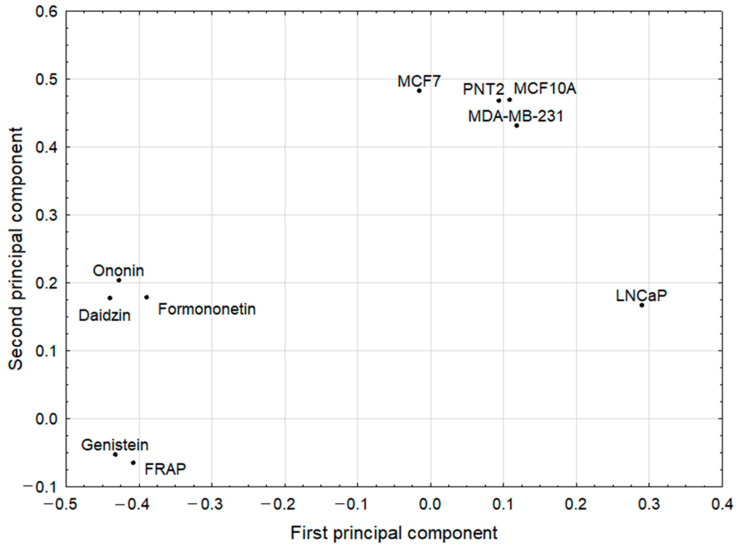
The variable loadings on the first and second principal components in the PCA model.

**Figure 5 pharmaceuticals-15-00806-f005:**
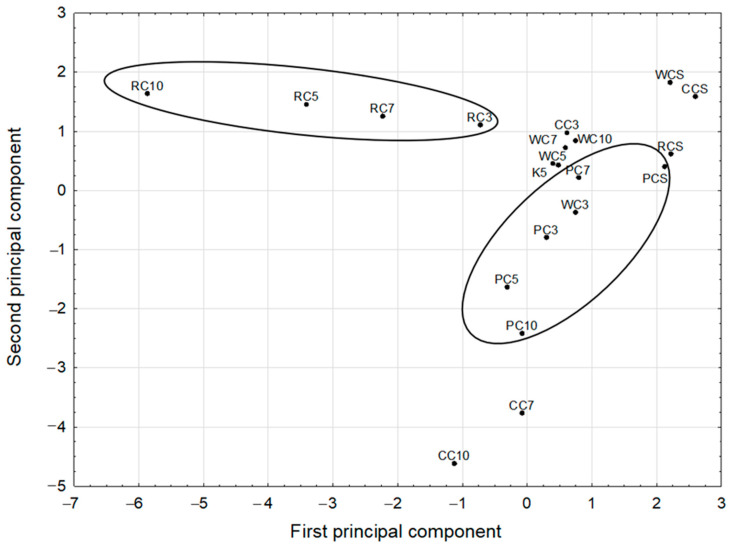
The projection of various species of clover sprouts into the space defined by the first two principal components of the PCA model.

**Table 1 pharmaceuticals-15-00806-t001:** Concentration of isoflavones and antioxidant activity of red (RC), white (WC), Persian (PC), and crimson (CC) clover seeds (S) and sprouts harvested for 3, 5, 7, and 10 days.

RCS	RC3	RC5	RC7	RC10	WCS	WC3	WC5	WC7	WC10	PCS	PC3	PC5	PC7	PC10	CCS	CC3	CC5	CC7	CC10
**ISOFLAVONE GLYCOSIDES [mg/100 g dw, *n* = 3]**																			
**daidzin**																			
Tr	2.81 ± 0.08	8.46 ± 0.12	5.52 ± 0.40	18.58 ± 1.40	Tr	Tr	Tr	Tr	Tr	Tr	1.47 ± 0.42	1.22 ± 0.05	0.99 ± 0.13	0.60 ± 0.08	Tr	Tr	Tr	Tr	Tr
**ononin**																			
Tr	193 ± 6	191 ± 3	229 ± 5	332 ± 10	Tr	11.87 ± 0.32	15.24 ± 0.35	19.93 ± 0.20	Tr	Tr	37.91 ± 4.48	32.18 ± 5.68	37.22 ± 0.64	21.95 ± 0.38	Tr	Tr	Tr	Tr	Tr
**ISOFLAVONE AGLYCONES [mg/100 g dw, *n* = 3]**																			
**daidzein**																			
Tr	Tr	Tr	Tr	Tr	Tr	Tr	Tr	Tr	Tr	0.13 ± 0.00	1.62 ± 0.20	1.16 ± 0.05	1.03 ± 0.06	0.68 ± 0.03	Tr	Tr	Tr	Tr	Tr
**formononetin**																			
Tr	5.97 ± 0.04	77.69 ± 2.39	34.96 ± 0.76	60.45 ± 1.39	Tr	27.12 ± 1.60	27.68 ± 0.68	28.19 ± 0.86	12.92 ± 0.39	Tr	3.79 ± 0.36	3.37 ± 0.35	3.14 ± 0.64	1.56 ± 0.04	0.07 ± 0.03	19.80 ± 1.98	14.82 ± 2.48	11.57 ± 0.44	7.98 ± 0.42
**genistein**																			
Tr	2.10 ± 0.14	6.00 ± 0.15	7.98 ± 0.20	15.12 ± 0.32	Tr	Tr	Tr	Tr	Tr	0.13 ± 0.01	2.42 ± 0.10	1.28 ± 0.08	0.94 ± 0.07	0.57 ± 0.04	0.07 ± 0.03	3.51 ± 0.62	2.90 ± 0.73	3.75 ± 0.89	9.75 ± 1.25
**ANTIOXIDANT ACTIVITY FRAP [** **µM/ Fe^2 +^ /g dw, *n* = 3]**																			
1.23 ± 0.16	44.60 ± 2.71	63.50 ± 1.30	33.87 ± 1.60	65.08 ± 3.16	1.17 ± 0.18	13.55 ± 0.92	29.53 ± 1.12	17.45 ± 0.78	31.54 ± 1.36	1.40 ± 0.17	26.00 ± 2.83	59.35 ± 2.06	28.44 ± 2.20	51.56 ± 1.62	1.03 ± 0.25	9.48 ± 1.02	23.47 ± 1.04	20.33 ± 0.96	29.10 ± 1.28
**ANTIOXIDANT ACTIVITY DPPH [µM Trolox/g dw, *n* = 3]**																			
0.29 ± 0.02	5.85 ± 0.24	9.65 ± 0.99	6.98 ± 0.31	13.88 ± 1.30	0.54 ± 0.10	17.88 ± 1.02	24.92 ± 1.26	11.65 ± 0.96	25.74 ± 1.57	0.32 ± 0.05	3.05 ± 0.22	5.05 ± 0.22	3.60 ± 0.29	7.28 ± 0.32	0.24 ± 0.09	2.04 ± 0.09	3.09 ± 0.16	2.78 ± 0.17	4.11 ± 0.14

Tr—traces; significant differences are shown in Appendix A.

**Table 2 pharmaceuticals-15-00806-t002:** Cytotoxic activity of the extracts of red (RC), white (WC), Persian (PC), and crimson (CC) clover seeds (S) and sprouts harvested for 3, 5, 7, and 10 days to breast (MCF10A) and prostate (PNT2) normal and cancer (MCF7, MDA-MB-231, and DU145, PC3, LNCaP, respectively) cells, expressed as IC_50_ values (µg/mL).

RCS	RC3	RC5	RC7	RC10	WCS	WC3	WC5	WC7	WC10	PCS	PC3	PC5	PC7	PC10	CCS	CC3	CC5	CC7	CC10
**PANEL OF BREAST CELLS**																			
**MCF10A**																			
↑	↑	↑	↑	↑	↑	↑	↑	↑	↑	↑	↑	↑	↑	440.6	↑	↑	↑	331.4	315.7
**MCF7**																			
↑	↑	↑	↑	↑	↑	352.2	↑	↑	↑	352.2	153.4	61.1	361.3	71.3	↑	↑	↑	58.9	61.1
**MDA-MB-231**																			
↑	↑	↑	↑	↑	↑	457.5	↑	↑	↑	↑	↑	↑	↑	412.9	↑	↑	↑	224.9	56.7
**PANEL OF PROSTATE CELLS**																			
**PNT2**																			
↑	↑	↑	↑	↑	↑	↑	↑	↑	↑	↑	↑	315	↑	429.1	↑	↑	481.4	237.9	119.5
**DU145**																			
430.5	↑	482.3	↑	488.6	↑	↑	↑	401.9	355	↑	117.9	↑	↑	261.1	↑	↑	↑	190.3	176.4
**PC3**																			
226.7	269.8	248.9	243.6	206.3	297.6	293.4	351.7	298.1	286.5	↑	203.1	324.9	415.9	187.6	↑	272.2	↑	72.2	32.9
**LNCaP**																			
↑	192.7	263.5	165.3	190.2	↑	354.5	260.7	189.3	239.4	↑	70.6	83.4	213.7	188.1	↑	37.6	95.8	234.0	153.8

↑ >Cmax.

## Data Availability

Data is contained within the article and Appendix A.

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
