# Peer review of "In the Search for Novel, Isoflavone-Rich Functional Foods—Comparative Studies of Four Clover Species Sprouts and Their Chemopreventive Potential for Breast and Prostate Cancer"

_pharmaceuticals, 2022, doi:10.3390/ph15070806_

Round 1

Reviewer 1 Report

This manuscript discovered the differences in isoflavones profiles among the four clover species sprouts: red, white, crimson, and Persian clover sprouts, cultured for different periods of time, and their impact on hormone-dependent cancers in vitro.

A few concerns about the manuscript for the authors to consider.

11.       In Figure 2, with the highest concentration tested of 500 µg/mL for each sample, it looks like the viability of the normal epithelial breast MCF10A cells were affected more than either of the two breast cancer cells, like WC10, PC3, PC5, PC7 etc. While the authors claimed that the sprouts tested were characterized by high safety for normal breast cells and decreased their viability only at the higher concentrations tested (>300 µg/mL) to 50-60%, it is not consistent with the data shown in Figure 2 when the concentration is higher than 300 µg/mL. Some of the sprout samples looked more toxic to the normal breast cell than to the breast cancer cells.

22.       It is interesting that the authors focused on isoflavone components in the tested samples from the title to Session 2.2 Different pattern in isoflavones accumulation dynamics among the tested sprouts, but the bioassay data did not support their initial hypothesis and it was apparent that the observed cytotoxic impact of the tested sprouts was not associated with isoflavones but probably with other compounds. Better organization/explanation/introduction may be helpful.

33.       Also for the viability of prostate cancer and normal cells, in Figure 3, some of the tested samples looked more toxic to the normal prostate epithelial cells PNT2 than to the prostate cancer cells. And the cytotoxic impact of the extracts tested did not correlate with the isoflavone content.

44.     In session 2.4. Antioxidant potential of clover sprouts not always corresponds with isoflavones amount, if the antioxidant properties of the sprout samples were responsible for the cytotoxic effects observed for breast cancer cells and prostate cancer cells, how can the authors explain the findings that Hormone-insensitive MDA-MB-231 breast cancer cells were more resistant to the tested extracts than estrogen-dependent MCF7 breast cancer cells, and androgen-dependent LNCap cells were most susceptible to the tested sprouts, followed by androgen-insensitive, high metastatic PC3 and low metastatic DU145 cells?

55.      In Figure 2, “WC” should be “WCS”.

Author Response

Reviewer#1

This manuscript discovered the differences in isoflavones profiles among the four clover species sprouts: red, white, crimson, and Persian clover sprouts, cultured for different periods of time, and their impact on hormone-dependent cancers in vitro.

A few concerns about the manuscript for the authors to consider.

In Figure 2, with the highest concentration tested of 500 µg/mL for each sample, it looks like the viability of the normal epithelial breast MCF10A cells were affected more than either of the two breast cancer cells, like WC10, PC3, PC5, PC7 etc. While the authors claimed that the sprouts tested were characterized by high safety for normal breast cells and decreased their viability only at the higher concentrations tested (>300 µg/mL) to 50-60%, it is not consistent with the data shown in Figure 2 when the concentration is higher than 300 µg/mL. Some of the sprout samples looked more toxic to the normal breast cell than to the breast cancer cells.

Answer: Thank you for this comment. We have demonstrated that the viability of normal breast cells was only affected in the concentration range between 300 and 500 µg/mL, what could be also observed in the IC50 values presented in Table 2. The IC50 values for normal breast cells in most cases exceeded the highest concentration tested of 500 µg/mL, and only for three samples the values were 315.7 – 440.6 µg/mL, while the IC50 values for the most vulnerable MCF7 cancer cells were in the range 61.1 – 361.3 µg/mL. In our opinion such results indicate that at the concentrations active to cancer cells, our samples were safe to normal breast cells. Unfortunately, at the highest concentration tested, for which the graphical visualization is presented on Figure 2, such differences are maybe not so convincing. We have modified the sentence describing the effect, to be more precise:

The most important thing is that at the concentrations cytotoxic to cancer cells, the sprouts tested were characterized by high safety for normal breast cells and the decrease in their viability was observed only at the higher concentrations tested (300 – 500 µg/mL).

It is interesting that the authors focused on isoflavone components in the tested samples from the title to Session 2.2 Different pattern in isoflavones accumulation dynamics among the tested sprouts, but the bioassay data did not support their initial hypothesis and it was apparent that the observed cytotoxic impact of the tested sprouts was not associated with isoflavones but probably with other compounds. Better organization/explanation/introduction may be helpful.

Answer: Thank you for this comment. We focused on isoflavones, as it was our initial idea to determine the content of the examined clover sprouts. We wanted to check both the level and accumulation dynamics of isoflavones, and also verify if the impact of the tested extracts, with the defined isoflavone content, on cancer cells is somehow related to their content. Cytotoxic analysis was the final step of our experiment, followed by statistical and chemometric analysis, which at the end indicated that the cytotoxic impact is not associated with the isoflavones content. Thus, the only thing we could do was to speculate what other substances might be responsible for the effect. On the basis of the preliminary qualitative HPLC analysis of phenolic compounds other than isoflavones we can speculate that it could be related to the presence of chlorogenic or gallic acid, observed in the results of the qualitative analysis, and such explanation is present in the text in paragraphs 2.3.1 (lines 185-189) and 2.3.2. (lines 227-229). However, the full quantitative analysis should be performed, followed by chemometric analysis, to explain which compounds are responsible for the cytotoxic impact. Another possibility is that there is a kind of synergistic effect between the compounds present in the extracts, resulting in the observed effect.

Also for the viability of prostate cancer and normal cells, in Figure 3, some of the tested samples looked more toxic to the normal prostate epithelial cells PNT2 than to the prostate cancer cells. And the cytotoxic impact of the extracts tested did not correlate with the isoflavone content.

Answer: The answer for this comment is similar to what we have wrote in the comment concerning the breast cancer and normal cells. In the case of normal prostate cells, the IC50 values for most of the samples exceeded the highest tested concentrations, and for a few samples the values were in the range of 119.5 – 481.4 µg/mL. Thus, with the exception of the CC10 samples, with IC50 119.5 µg/mL, the rest of the IC50 values indicate high safety of the tested samples, at the concentrations cytotoxic to cancer cells. We have changed the sentence to be more precise:

Interestingly, the effect observed for normal prostate cells was very weak, with the lowest IC50 of 119.5 and 237.9 µg/mL only for CC10 and CC7 sprouts, respectively, what suggests that at the doses cytotoxic to cancer cells the tested extracts (with the exception of CC10 and CC7 samples) were safe to normal cells.

In session 2.4. Antioxidant potential of clover sprouts not always corresponds with isoflavones amount, if the antioxidant properties of the sprout samples were responsible for the cytotoxic effects observed for breast cancer cells and prostate cancer cells, how can the authors explain the findings that Hormone-insensitive MDA-MB-231 breast cancer cells were more resistant to the tested extracts than estrogen-dependent MCF7 breast cancer cells, and androgen-dependent LNCap cells were most susceptible to the tested sprouts, followed by androgen-insensitive, high metastatic PC3 and low metastatic DU145 cells?

Answer: We do not hypothesize that the observed cytotoxic effects result from the antioxidant potential of the samples. None of the correlation weights for the pairs of parameters calculated in our PCA model indicated such relationship. The presented results are only preliminary, thus right now we cannot predict the potential mechanism of cytotoxic impact of the tested samples on cancer cells. The only thing we can observe at this stage of the study is a kind of similarity in the response of both hormone sensitive cell lines, namely MCF7 and LNCaP, to the examined samples – both cell lines were more susceptible than the hormone independent cells used in the study. However, the explanation of this phenomenon needs further studies, as isoflavones was not the active group that decreased cell viability. Maybe there is a synergistic effect of isoflavones and other phenolics present in the extracts, but right now this is only a speculation, which needs further evidence.  

In Figure 2, “WC” should be “WCS”.

Answer: Thank you for this comment, this was corrected, both in Figure 2 and 3.

Reviewer 2 Report

I am somewhat concerned about the idea in this manuscript. In chemotherapy, it is very important that a sufficient dose is given to prevent the development of resistance. In this study residue of extracts with IC50 values of 200 – 500 µg/ml are suggested as functional foods for cancer patients. The authors have not calculated how much of the residue of the extract is needed to get a concentration near the cancer cells exceeding the IC50 value if taken orally. Neither have the authors calculated how much of the sprouts must be eaten to get a pharmacologically active dose of isoflavanoids. I fear that such a calculation would reveal that an exceedingly high amount would have to be eaten. As a consequence, you could argue that eating these sprouts would facilitate the development of resistance in cancer patients since a too low dose is given.

I am also concerned about the reproducibility of the assays. Apparently, only one experiment has been performed in triplicate. A result of a biologic assay only performed once even in triplicate can not be published.

The abbreviations WC, RC, Pc and CC are only explained in the legend to Table 1.

The abbreviations PCS and CCS are not explained.

A number like 193.25+-6.01 should just be 193+-6 to reveal the importance of the standard deviation.

I do not see a sum of 426 mg/100 g anywhere in table 1. If I add the numbers 332+60+15 = 407.

It is explained the extract is used for cytotoxic studies. However, is the residue of the concentrated extract meant? See e.g. legend to Table 2: cytotoxicity of ..  seeds. How can you incubate cells with seeds?

Author Response

Reviewer#2

I am somewhat concerned about the idea in this manuscript. In chemotherapy, it is very important that a sufficient dose is given to prevent the development of resistance. In this study residue of extracts with IC50 values of 200 – 500 µg/ml are suggested as functional foods for cancer patients. The authors have not calculated how much of the residue of the extract is needed to get a concentration near the cancer cells exceeding the IC50 value if taken orally. Neither have the authors calculated how much of the sprouts must be eaten to get a pharmacologically active dose of isoflavanoids. I fear that such a calculation would reveal that an exceedingly high amount would have to be eaten. As a consequence, you could argue that eating these sprouts would facilitate the development of resistance in cancer patients since a too low dose is given.

Answer: Thank you for this interesting and valuable comment. The idea of our study was to search for the functionals foods of rather chemopreventive properties. Thus, we would like to emphasize that the use of such products by patients with so far existing or developed cancer disease is not advisable, as the hormonal balance in the organism with hormone-dependent cancer can be easily disturbed by the uncontrolled intake of isoflavones with food products. We only wanted to point out what are the amounts of isoflavones in the clover sprouts, as some of them are commercially available, without any information of the isoflavones content. In the conclusion part we have calculated that the amount of the sum of isoflavones for 10-day red clover sprouts in their fresh form is about 20 mg/100 g, which is the amount present in some dietary supplements for menopause. The data from some recent analysis concerning the risk of breast cancer and the isoflavones intake with foods indicate that the amounts of 36 to over 230 mg of isoflavones per day did not increased the cancer risk (Finkeldey, L.et al., Effect of the Intake of Isoflavones on Risk Factors of Breast Cancer—A Systematic Review of Randomized Controlled Intervention Studies. Nutrients 2021, 13, 2309.  https://doi.org10.3390/nu13072309), however the final conclusions should be carefully drawn and more studies are still needed.  

I am also concerned about the reproducibility of the assays. Apparently, only one experiment has been performed in triplicate. A result of a biologic assay only performed once even in triplicate can not be published.

Answer: Thank you for this comment, we completely agree that the biological assays should not be performed as single experiment. Our data comes from three independent experiments, that is what we understand as “triplicate”. The statements were corrected throughout the text.

The abbreviations WC, RC, Pc and CC are only explained in the legend to Table 1.

Answer: The abbreviations are explained in the legends for each table and figure

The abbreviations PCS and CCS are not explained.

Answer: PCS and CCS are the seeds of Persian clover and crimson clover. The abbreviations for PCS and CCS, together with all other abbreviations used throughout the text, are explained in paragraph 4.2. Moreover, the abbreviations are also explained in the legends to tables and figures.

A number like 193.25+-6.01 should just be 193+-6 to reveal the importance of the standard deviation.

Answer: We have presented our results in the form of the mostly used standard format of the mathematical notation, including also the decimal values.

I do not see a sum of 426 mg/100 g anywhere in table 1. If I add the numbers 332+60+15 = 407.

Answer: Thank you for this observation, in Table 1 we have missed one value of daidzin amount in RC10 sprouts, which is 18.58±1.4. We have corrected this and now the sum is 426 mg/100 g.

It is explained the extract is used for cytotoxic studies. However, is the residue of the concentrated extract meant? See e.g. legend to Table 2: cytotoxicity of ..  seeds. How can you incubate cells with seeds?

Answer: Thank you for this comment. Yes, we have used the residues of the concentrated extracts for the cytotoxic studies. We have corrected the explanation in the legends for Table 2 and Figures 2 and 3, as the expression “cytotoxicity of …. the seeds” was the obvious mental shortcut.

Reviewer 3 Report

Manuscript Number: Pharmaceuticals-1789287

entitled: In the search for novel, isoflavone-rich functional foods – comparative studies of four clover species sprouts and their chemo-preventive potential for breast and prostate cancer

The authors conducted important data. This is an interesting paper. The present version of the manuscript is very well developed and the manuscript is well written. The data presented are new and relevant. The present version is much better; therefore, the manuscript is suitable for publication in its current form.

Author Response

Reviewer#3

The authors conducted important data. This is an interesting paper. The present version of the manuscript is very well developed and the manuscript is well written. The data presented are new and relevant. The present version is much better; therefore, the manuscript is suitable for publication in its current form.

Answer: Thank you very much for your kind opinion and recommendation.

Round 2

Reviewer 2 Report

I now understand the authors want to make the point, that they do not think that the food can cure cancer, but instead might increase the risk of developing hormone-resistant cancer. I appreciate their investigation to reveal how much isoflavonoid you in reality consume. However, I still think that the authors in the conclusion or somewhere in the introduction should make it clear that they fear those functional foods with isoflavonoids may be a problem for persons carrying hormone-sensitive cancer diseases. I do not know if scientific data support this hypothesis. In the present formulation line 55 I understand that they might improve the hormone treatment. I suggest the authors make their point clear, somewhat as in their reply to me.

I am happy that the authors have made their assays three times. I still believe that triplicate means that one experiment is formed with three vials at the same time.

It is true that the authors by using the term 193.25+-6.01 use common but unfortunate writing. The meaning of the term is that the correct number with a probability of 95% is between 190 and 196. Consequently, it makes no sense to write 193.25 +-6.01. I also have noticed that the ridiculous way of writing experimental numbers with too many important figures is now in common use. I would suggest editors stop the promotion of a notation that indicates that the authors do not know what they are doing. The number of figures should reveal the accuracy with which the result is obtained.

Author Response

Reviewer#2 round#2

I now understand the authors want to make the point, that they do not think that the food can cure cancer, but instead might increase the risk of developing hormone-resistant cancer. I appreciate their investigation to reveal how much isoflavonoid you in reality consume. However, I still think that the authors in the conclusion or somewhere in the introduction should make it clear that they fear those functional foods with isoflavonoids may be a problem for persons carrying hormone-sensitive cancer diseases. I do not know if scientific data support this hypothesis. In the present formulation line 55 I understand that they might improve the hormone treatment. I suggest the authors make their point clear, somewhat as in their reply to me.

Answer: Thank you for this comments and for the appreciation of our work. We have added the following statements in the Introduction:

Sprouts rich in phytoestrogens (including isoflavones) are an example of  functional food that may reveal chemopreventive potential against the development of hormone-dependent cancers. However, as the isoflavones provided with the diet can influence the hormonal balance in the human body, it is crucial to know what is their content in a given product. This is especially important for people with the risk of the development or already existing hormone-dependent cancers.

and also in the Conclusions:

Thus, special caution should be undertaken before including clover sprouts in the diet of the consumers with the risk of hormone-dependent cancers. Moreover, the women taking isoflavone-rich dietary supplements for reducing their menopausal symptoms should also be aware that some clover sprouts can deliver additional amount of isoflavones.

I am happy that the authors have made their assays three times. I still believe that triplicate means that one experiment is formed with three vials at the same time.

Answer: Thank you for your comment.

It is true that the authors by using the term 193.25+-6.01 use common but unfortunate writing. The meaning of the term is that the correct number with a probability of 95% is between 190 and 196. Consequently, it makes no sense to write 193.25 +-6.01. I also have noticed that the ridiculous way of writing experimental numbers with too many important figures is now in common use. I would suggest editors stop the promotion of a notation that indicates that the authors do not know what they are doing. The number of figures should reveal the accuracy with which the result is obtained.

Answer: Thank you for the explanation, we have corrected the notations.